# Chlorophyll Meter: A Precision Agricultural Decision-Making Tool for Nutrient Supply in Durum Wheat (*Triticum turgidum* L.) Cultivation under Drought Conditions

**DOI:** 10.3390/life13030824

**Published:** 2023-03-17

**Authors:** Anteneh Agezew Melash, Bekir Bytyqi, Muhoja Sylivester Nyandi, Attila Miklós Vad, Éva Babett Ábrahám

**Affiliations:** 1Kálmán Kerpely Doctoral School of Crop Production and Horticultural Science, University of Debrecen, Böszörményi Street 138, H-4032 Debrecen, Hungary; 2Department of Horticulture, College of Agriculture and Environmental Science, Debark University, North Gondar, Debark P.O. Box 90, Ethiopia; 3Institutes for Agricultural Research and Educational Farm, University of Debrecen, H-4032 Debrecen, Hungary; 4Institute of Crop Sciences, Faculty of Agricultural, Food Sciences and Environmental Management, University of Debrecen, Böszörményi Street 138, H-4032 Debrecen, Hungary

**Keywords:** chlorophyll meter, durum wheat, leaf reflectance traits, drought, nutrient supply, yield, protein content

## Abstract

How crop biodiversity adapts to drought conditions and enhances grain yield became the most important issue facing agronomists and plant breeders at the turn of the century. Variations in genetic response, inadequacy of nutrients in the soil, and insufficient access to nutrients are factors that aggravate drought stressors. The development of screening tools for identifying drought tolerance is important in the deployment of durum wheat varieties suited to drought-prone environments. An experiment was conducted to evaluate durum wheat varieties under a range of nutrient supplies in naturally imposed drought conditions. The treatments consisting of two nitrogen regimes (i.e., control and 60 kg ha^−1^), four durum wheat varieties, and three types of nutrients (control, sulfur, and zinc) that were arranged in a split-split plot design with three replications. Both foliar-based sulfur and zinc fertilization were employed at the flag leaf stage, at a rate of 4 and 3-L ha^−1^, respectively. The results showed a significant (*p* < 0.05) genetic variation in chlorophyll concentration, grain protein content, tillering potential, and leaf area index. Varieties that contained better leaf chlorophyll content had improved grain yield by about 8.33% under 60 kg/ha nitrogen. A combined application of nitrogen and zinc at flag leaf stage significantly improved grain yield of Duragold by about 21.3%. Leaf chlorophyll content was found to be a more important trait than spikes per m^2^ to discriminate durum wheat varieties. Foliar application of sulfur increased the grain yield of drought-stressed plants by about 12.23%. Grain yield and protein content were strongly correlated with late-season SPAD readings. Significant (*p* < 0.05) correlation coefficients were obtained between normalized difference vegetation index, leaf area index, grain yield, and protein content with late-season chlorophyll content, revealing the importance of chlorophyll content in studying and identifying drought-tolerant varieties.

## 1. Introduction

Durum wheat *(Triticum turgidum* L.) grain is an important ingredient for making different food products such as pasta, couscous, and bulgur, which provides about 20% of the global protein and calories consumption [1]. A proliferation of the world population and expansion of food industries creates an opportunity to further improve durum wheat productivity while concomitantly safeguarding the environment. Indeed, harvesting high grain yield is the outcome of several physiological processes and associated traits, such as chlorophyll content in photosynthetic organs [2]. However, these important plant features are highly depending on nutrient application and management [2,3]. Any form of nutrient application during the crop growing period is directly related to the photosynthetic function of the leaves [4]. Nutrients such as nitrogen regulate leaf photosynthetic pigment synthesis, and the level at which nitrogen is applied to durum wheat has a significant positive correlation with the chlorophyll content [5]. However, nitrogen application at a higher rate could reduce grain yield mainly through reducing the rate of grain filling of the middle spikelets, which substantiates proper nitrogen management [6]. This implies that the use of nitrogen either in excess or insufficient doses could be a potential cause of yield reduction in durum wheat. Hence, effective and site-specific nutrient management, monitoring, evaluation, and regular diagnosis of plant nutrient status during the growing season is very important as far as the environment and productivity is concerned.

Nutrient status and photosynthetic machineries such as leaf chlorophyll concentration have been previously measured using conventional methods, which includes a complex procedure of solvent extraction followed by in vitro spectrophotometric determination [7]. This assumption shows the need for alternative and cost-effective methods for a rapid crop nutrient status analysis in the durum wheat cropping system. Modern precision agricultural tools such as chlorophyll meters (i.e., SPAD) allow instant real-time, rapid, and non-destructive assessment of chlorophyll content and crop nitrogen status [8,9]. These measurements can enable decision makers to further refine nitrogen recommendations for divergent pedoclimatic conditions [8,10]. A wealth of information is available, highlighting the use of SPAD measurements as a crop nitrogen status monitor for grain yield prediction and nitrogen utilization efficiency assessments [11,12]. Hence, as the leaf chlorophyll concentration is highly correlated to nitrogen status [13], the leaf spectral characteristics could be used to guide nitrogen fertilizer use [12].

Although SPAD-based nutrient management offers several benefits, it has been found to be more sensitive to varietal differences and interactions among nutrients [14], pedoclimatic factors, biotic stress [15], growth stage, and seasonal variability [16]. These factors significantly influence efficient utilization of nitrogen, a fundamental constituent of leaf chlorophyll content [17]. It is stated in the literature that restrictions in sulfur acquisition have been observed as a primary cause for reduced growth and grain yield, particularly under water-limited environments [18]. Hence, under inadequate fertilization of sulfur, wheat varieties cannot exploit their full yield potential, and the crop may not efficiently use the applied nutrients [19]. Nitrogen also influences sulfur availability due to their strong synergetic association, and since both are involved in protein synthesis, deficiency in nitrogen can also cause sulfur deficiency in the grain [20,21]. This means that sulfur deficiency can decrease nitrogen utilization efficiency and that nitrogen deficiency could also reduce sulfur use efficiency [22]. These results universally imply an absolute requirement of a balanced N:S ratio to obtain high yields and acceptable grain quality [23]. As such, there is a reduction in SPAD readings due to an imbalance between nitrogen and sulfur nutrients [24]. In view of these facts, this study aimed to (i) evaluate the response of leaf chlorophyll content in durum wheat to changes in varying applications of nutrients over a complete growth cycle, and (ii) to appraise the effect of nitrogen, zinc, and sulfur fertilizer application on leaf chlorophyll content, grain yield, and protein content of durum wheat varieties.

## 2. Materials and Methods

### 2.1. Genetic Materials and Study Site Description

This study used four genetically diverse and promising durum wheat varieties (i.e., Colliodur, Durablank, Duragold, and Tamadur) sourced from Austria and Slovakia. The varieties such as Duragold and Durablank are mid-late/late maturing varieties, and known for their high yielding potential, tillering capacity, high stability, high agronomic responsiveness, and good grain quality (https://istropol.sk/hu/katalogus, accessed on 6 March 2023). The Tamadur variety is also characterized by its good agronomic characteristics, high grain quality traits, and resistance to multiple diseases, while Colliodur is known for its high yielding capacity and high drought tolerance level. These vegetal materials were then evaluated during the spring season at the Látókép research center of the University of Debrecen, Hungary, which is located at 47°33042″ N; 21°27002″ E, about 15 km from Debrecen [25]. This site is characterized by calcareous chernozem soil which contains about 2.7–2.8% humus, a nearly neutral pH (pH KCl = 6.46), and a specific plasticity index (KA) of 40 [26]. Soil samples were analyzed and indicated that the soil of the experimental station contained about 57 mg kg^−1^ AL-soluble phosphorus, 199 mg kg^−1^ AL-soluble potassium, and 0.94 mg kg^−1^ of zinc. The water management characteristics of the soil were favorable. Soil samples were taken from 0–20 cm at different locations prior to application of nutrients during the experimental season. The daily average temperature, relative air humidity and seasonal precipitation during the test period (August 2020–June 2021) are shown in Figure 1. The collected meteorological data for the 2020/21 cropping season clearly showed that there was severe drought stress, particularly at the beginning of the flowering and maturity periods. The growing season was warmer and dryer than the average temperature and precipitation of the 30-year period (Figure 1).

### 2.2. Experimental Design and Treatments

The four durum wheat varieties were combined with three foliar-based fertilizers (i.e., control, sulfur, and zinc) and two levels of nitrogen application regimes (i.e., without and 60 kg/ha) were arranged in a split-split plot design with three replications. The main plots were assigned to the two levels of nitrogen application rates, the subplots to the four durum wheat varieties, and the sub-sub plots to the three levels of foliar-based nutrient application. The plot size of the sub-sub plots was 9 × 1.5 (13.5 m^2^) with a 15 cm row spacing. The full plot size was used to measure grain yield as kg of grain produced per plot, then converted into t/ha. Both zinc and sulfur fertilizers were applied at the flag leaf stage at a rate of 3 and 4 L ha^−1^, respectively. Other agronomic practices such as pest and disease control measures were applied uniformly for all experimental plots as recommended for wheat. Throughout the growing period, the experimental site was characterized by dry climatic conditions, with an average monthly rainfall of 41 mm and high inter-annual and seasonal variability (Figure 1).

### 2.3. Data Collection

Chlorophyll content (*SPAD readings*): A hand-held chlorophyll meter (Minolta SPAD-502) was used to measure the relative leaf chlorophyll content and crop nitrogen status. This reflectance-based trait was non-destructively measured every two weeks from the 10 uppermost fully expanded and intact leaves starting from 65 to 128 days after sowing (DAS) and onwards. To prevent interference from external light, the chlorophyll meter was clamped onto a single leaf. This measurement was made across all replication and experimental units uniformly. The instrument measures leaf transmittance centered at red light (650 nm; peak chlorophyll absorbance) and NIR (940 nm; non-chlorophyll absorbance) wavelengths. At these points of transmission, the Minolta SPAD-502 m calculates the amount of relative leaf chlorophyll content of the leaves, according to Equation (1) [27].
(1)$$SPAD\;readings=A\times \left[\log\;\left(\frac{Ior}{Ir}\right)-\log\;\left(\frac{Iof}{If}\right)+B\right]$$
where:*A* = constant;*B* = constant;*Ior* = current from red detectors with sample in place;*Ir* = current from red infrared detectors with sample in place;*Iof* = currents from red detectors with no sample;*If* = currents from infrared detectors with no sample

Grain yield: was determined in a 13.5 m^2^ (1.5 m × 9 m) area by a Sampo Rosenlew SR 2010 plot combine harvester equipped with a Coleman weighing system and was recorded as kg/ha, which is then converted into t/ha after adjusting the harvested grain to account for the 12.5% moisture content, as described by Badu-Apraku et al. [28].
Grain yield (kg ha^−1^) = (100 − % AMC)**/**(100 − % SMC) × 100(2)
where:AMC = Actual (obtained) Grain Moisture Content (%)SMC = Standard Moisture Content

Leaf area index (LAI), Normalized Difference Vegetation Index (NDVI), and Spike density (m^2^): We took measurements of LAI and NDVI from 8 to 10 AM every two weeks, at different developmental stages (at 65, 77, 92, 128 DAS). LAI was measured at four different developmental stages (65–128 DAS) using a Delta-T SunScan SS1 COM-R4 portable plant canopy analyzer system with a radio link (Delta-T Devices Ltd., UK). This device records light transmissions, analyzes the incident and transmitted PAR (photosynthetically active radiation) of the durum wheat canopy with a 100 cm long probe that has 64 PAR sensors with a spectral range of 400–700 nm. The readings are based on units of photosynthetically active radiation quantum flux (µmol m^−^^2^ s^−^^1^) and units of leaf area index (m^2^ m^−^^2^). The NDVI was also recorded every two weeks using a portable N tech “Trimble Greenseeker” NDVI meter at various growth stages of the crop following Equation (3) [29]. This device was positioned at 1 m over the durum wheat canopy, and readings were obtained for the average value of the entire plot. At the maturity stage, we also measured the spike density using a 1 m square (1 × 1 m) quadrant and counting the number of productive spikes produced within the quadrant.
NDVI = (NIR − RED)**/**(NIR + RED) (3)

Protein and moisture content: A machine called a Pfeuffer Granolyser NIR (Pfeuffer, Germany) was used to determine grain protein and moisture content. It uses NIR (near infrared) diode technology, making 1500 individual scans per sample. The built-in spectrometer scans the sample seeds within the range of 950 to 1540 nm.

### 2.4. Statistical Data Analysis

The results are expressed as mean values and the error bars represent the standard error (SE) of the mean. Prior to statistical assessments, the normality and homogeneity of the data were tested by the Shapiro–Wilk test (Shapiro and Wilk, 1965) [30]. The majority of the studied traits had normally distributed values. The statistical data analysis was carried out using the GenStat (18th ed.) statistical software package [31]. When the analysis of variance (ANOVA) revealed a significant difference, the means of the treatments were compared using the least significance difference (LSD) *t*-test (*p* ≤ 0.05) to determine the differences between expression levels of nutrients under drought-stressed conditions. We compared the interaction of the treatments presented in different graphs and tables using Duncan’s Multiple Range Test (DMRT). The graphs were constructed by MS Excel graphical functions. Paired Spearman’s correlation analysis was also conducted using the GenStat (18th ed.) statistical software package to ascertain the magnitude and strength of the association between pairs of traits under the applied nutrients.

## 3. Results

### 3.1. Genetic Regulation and Stability of Chlorophyll Content

The SPAD readings, estimated by a chlorophyll meter, were significantly (*p* < 0.05) affected by the genetic landscape of the tested varieties (Table 1). These varieties had a remarkable ability to sense environmental factors such as nutrient supply and drought stress, which revealed an array of substantial genetic variations in relative leaf chlorophyll concentration (Table 1, Table 2 and Table 3). Since the outset of yellowing and greening of the leaves determined by the stay green nature of durum wheat varieties, the corresponding SPAD values ranged from 54.3 to 62.7 and 49.0 to 52.3 for Durablank and Colliodur, respectively (Table 2). This shows the slow degradation of the leaf chlorophyll concentration for the Durablank variety under drought conditions, which could be due to its drought tolerance and high responsiveness to nitrogen fertilization. The varieties that contained the optimum leaf chlorophyll content could further maintain the maximum photosynthetic efficiency, which enabled them to achieve maximum yield formation.

### 3.2. Interaction of Durum Wheat Varieties and Nutrient Supply under Drought Conditions

The analysis of variance showed that exogenous application of nitrogen and sulfur fertilizers significantly (*p* < 0.05) enhanced grain yield during drought episodes (Table 1). The yield effect of sulfur was significantly varied in the extent to which the durum wheat varieties responded to drought-induced stress and responsiveness to the applied inputs (Figure 2). When the Tamadur variety was treated with sulfur-containing fertilizer, the grain yield was significantly improved by about 12.23% (4.8 to 5.4 t/ha) compared to the control (Figure 2).

### 3.3. Dynamics in SPAD Values (Chlorophyll) and Its Contribution to Grain Yield

The analysis of variance unveiled that the SPAD readings were significantly (*p* < 0.05) affected by the genetic landscape of the durum wheat varieties and the applied inputs (Table 1, Table 2, Table 3 and Table 4). When relative chlorophyll content was used to differentiate the crop’s biological potential, the varieties that contained a higher chlorophyll content achieved a higher grain yield (Table 1). The Duragold (7.5 t/ha) and Durablank (6.9 t/ha) varieties in this sense achieved potentially maximum grain yields, probably because of sufficient leaf chlorophyll contents (Table 1 and Table 2). Higher SPAD readings could be an indicator of whether the durum wheat varieties received an adequate amount of nitrogen for grain formation during the growing season. However, under 60 kg/ha of nitrogen application, better chlorophyll-content-producing varieties did not maintain higher grain yields (Table 3). This was explicitly observed in the two highest yielding durum wheat varieties; Duragold maintained higher leaf chlorophyll concentration than Durablank but the opposite occurred when this specific trait was used to differentiate between the yielding potential of both varieties (Table 2, Table 3 and Table 4). This means that, under drought conditions, both enhanced and decreased leaf chlorophyll content could affect the grain yield of crops.

Additionally, our result showed that, under drought stress conditions, leaf chlorophyll content was more important for grain yield than any other traits such as spike per square meter (Table 1 and Table 2). This had been observed in the Colliodur variety that had relatively lower grain yields (6.7 t/ha) despite its higher number of produced spikes per m^2^ (Table 2). These variations could be due to the conversion and allocation of the nitrogen, zinc, and sulfur fertilizers into grain formation and the pedoclimatic influence of the growing environment.

### 3.4. Evaluation of Durum Wheat Varities Nitrgen Status Using Chlorophyll Meters

Application of nitrogen at the flag leaf stage significantly (*p* < 0.05) influenced the SPAD readings; the extent at which nitrogen influenced the SPAD values varied between the varieties and developmental stages (Table 1, Figure 2). Although the leaf nitrogen status could be measured using the chlorophyll meter, divergence in varietal responses to the applied inputs and drought stress could influence the relationship between the SPAD value and nitrogen status. A SPAD reading of 53.3 to 60.1 for durum wheat, which represents a leaf area-based nitrogen concentration, was measured, with the highest value was found to be around 92 DAS (Figure 3). This range could be considered as a benchmark above which no further addition of nitrogen fertilizer would be necessary, especially if split application of nitrogen is employed. When the relationship between nitrogen and SPAD readings was examined, the SPAD value was more prominent at higher (60 kg/ha) nitrogen application rates than that in the standard check. This shows that the proper nitrogen application rate can improve the photosynthetic activity of durum wheat varieties under drought conditions (Table 2). The association between nitrogen and SPAD readings shows the potential to use the chlorophyll meter in predicting durum wheat nitrogen status, leaf chlorophyll content, and grain yield in the framework of the current climate change scenarios.

### 3.5. Developmental Stages Determine the SPAD Readings

A statistically significant (*p* < 0.05) repercussion of developmental plasticity was observed on leaf chlorophyll content (SPAD) (Figure 4). The response of leaf chlorophyll content to developmental stages was found to be quadratic (Table 2, Table 3 and Table 4). There was a rapid increase in SPAD values at the first stages (i.e., 62–92 DAS), while its value stagnated at 128 DAS onwards, even though nitrogen, zinc, and sulfur fertilizers were applied foliarly at the vegetative stages (Table 4). Such a quadratic pattern might be because of a rapid degradation of chlorophyll content and leaf discoloration, particularly at the later growth phases. A decline in chlorophyll content later in the growth stages under zinc and sulfur fertilization further indicates that applying sulfur- and zinc-containing fertilizers late in the growing stages may not have economical and agronomical benefits. These suggest that, under drought conditions, zinc and sulfur fertilizers should be applied early in the growth season rather than late in the growth stages to maintain optimum leaf chlorophyll concentration.

### 3.6. Interaction Effect of Nitrogen × Sulfur × Zinc on Spikes Density (m^2^)

In the tested environment, spike density was significantly (*p* < 0.05) influenced by application of nitrogen, sulfur, and zinc fertilizers (Table 2 and Table 3). Significant phenotypic variation for spike density (m^2^) was also observed among the durum wheat varieties. Despite its high yielding capacity, variety Duragold attained a lower (206) number of spikes per square meter (Figure 5). Meanwhile, the maximum (216) and minimum (154) number of spikes were observed for Colliodur and Tamadur, respectively (Table 2). These variations could be potentially correlated with their tillering potential and developmental plasticity, particularly when water becomes very limited in supply.

Nitrogen application and varietal difference were statistically significant (*p* < 0.05) for spike density (Table 3, Figure 5). Under nitrogen supply, differences among the varieties were greater than those with the unfertilized soil condition. Although the biological potential of the first two high yielder varieties was similar in the production of more spikes per square meter, Colliodur attained the highest spikes per m^2^ followed by Durablank at 60 kg N ha^−1^ (Table 3). This means the varieties presented distinct physiological response, which could be due to differences in nitrogen uptake and utilization efficiency. Additionally, a sufficient amount of nitrogen fertilization could improve the number of spikes per square meter, probably through promoting the tillering capacity of durum wheat. The lower number of spikes recorded for the Tamadur variety could be due to either its poor tillering potential or non-responsiveness to the applied nitrogen fertilizers. This means that the formation of a substantial number of spikes (m^2^) could be a significant genetic gain of fertile tillers. Improving tillering potential of durum wheat varieties through promoting their resource utilization efficiency could therefore be a practical solution to enhance the spike density and yield.

### 3.7. Canopy Reflectance Sensor-Based Fertilizer Management

The application of nitrogen was efficient in creating variability in NDVI and LAI at the stage of six fully expanded leaves. The interaction effects among nitrogen × varieties × foliarly applied nutrients on the studied biophysical traits such as LAI and NDVI were statistically significant (Table 4). Co-fertilization of foliar-based sulfur-containing nutrients with 60 kg/ha of soil-based N fertilizer increased the LAI and NDVI profiles in a variety-dependent manner (Table 4; Figure 6). Throughout the growing season, LAI and NDVI ranged from 2.7 to 4.9 and 0.67 to 0.82, respectively, for the Colliodur variety (Figure 6). This indicated that these biophysical traits could be enhanced in a growth stage- and nutrient application rate-dependent manner to the extent that even the slightest adjustment in the nitrogen fertilization dose could change NDVI and LAI profiles during the growing season. Through the application of sulfur fertilizer, Durablank measured higher SPAD values than the other tested varieties (Table 4). These results clearly indicated that divergent varieties of the same plant species could respond completely differently to the applied nutrients, suggesting the need for suitable genotype selection coupled with a specific set of agronomic (nutrient management) packages in stressful environments.

NDVI and LAI were found to be biophysical traits that are sensitive to a change in developmental phases and nutrient supply (Figure 6 and Figure 7). As confirmed in the combined analysis of variance, both traits showed similar trends to increase and then decreased as the developmental phases progress. The maximum measurement of NDVI (0.84) (Figure 6) and LAI (55.5) were observed around 77 DAS, but stagnated afterwards (Figure 7). This result further confirmed that the LAI and NDVI values in durum wheat may not be similar during the leaf formation stage and later during maturity or senescence stages. This might be due to the fact that during early developmental stages, all leaves could green whereas in the latter stages, the color of the leaves became a blend of yellow, green, and dead leaves which decreased the LAI and NDVI measurements.

### 3.8. Nutrient Supply Influences Grain Protein Content (%)

Grain protein content was significantly (*p* < 0.05) influenced by all the imposed factors and their interactions as well (Table 2, Table 3 and Table 4). The deviation in grain protein content was significantly wider due to nitrogen application treatment than for zinc, sulfur, and genetic variation. This means that protein content is a grain qualitative trait that is more sensitive to nitrogen application than any other studied factors. When the sole effect of nitrogen is considered, the grain protein concentration was found to be higher in durum wheat varieties grown with an adequate nitrogen rate (60 kg/ha) compared to the control (Table 3). However, this assumption was found to be inconsistent due to variability in varietal response to the applied inputs. It was also observed to be higher in the Duragold variety sown under an optimum nitrogen rate of 60 kg/ha (Figure 8); this indicated that the grain protein content is under the control of the genetic landscape and physiological response of the varieties to the applied inputs. Hence, screening and revisiting the old, orphan, and landrace durum wheat varieties could be an important pool to reduce the tradeoff between yield and grain protein content, as durum wheat grain protein content matters in the food industries.

A strong positive and significant correlation between grain protein concentration and SPAD readings was observed across the developmental phases (Figure 9). It was also clearly observed that the varieties that contained higher leaf chlorophyll contents accumulated higher grain protein contents (Table 2). This association clearly showed that grain protein concentration can be estimated from leaf chlorophyll measurements and leaf nitrogen status. Hence, it is very important to study allometric associations between dynamics in grain protein content and leaf chlorophyll content, especially under resource-limited environments to further improve grain protein concentration. In fact, protein synthesis could be closely associated with the transport and assimilation of nitrogen metabolites and carbon into the grain. Across the varieties, the SPAD readings at the heading stage were between 53.5 and 62.3, and thus application of 60 kg N ha^−1^ was required to attain about 12% grain protein content.

### 3.9. Relationships between SPAD Readings, Yield, and Morphological Traits

The correlation coefficients for the studied physio-morphological traits and grain yield are presented in a graphical form for better visualization of the relationship and changes between SPAD, NDVI, and grain yield (Figure 9). With the progression of developmental plasticity, there was a strong association between NDVI and grain yield (Figure 9), evidencing the potential of this tool to assess the status of durum wheat and estimate grain formation under drought conditions. Although the strength of the correlation coefficient was dictated by crop developmental stages, the coefficient of determination for the relationship between SPAD values with grain yield was found to be medium. This indicated that the SPAD meter could be a potential agricultural device to analyze the spatial variability of the durum wheat under field conditions to estimate the grain yield throughout the growth cycle of the crop.

Additionally, a strong positive correlation between NDVI and spike density (m^2^) was observed (Figure 9). This means that an increase in the spike density per square meter could increase the seasonal NDVI profile (Figure 9). This increase could be strongly associated with the tillering potential of the varieties and additional decumbent leaves, resulting in substantial ground coverage. However, the correlation between NDVI and SPAD readings was weakly positive, particularly at the early growth stages. The grain yield was also significantly correlated with the NDVI values at which stronger associations were observed at the latter developmental stages.

## 4. Discussion

Sustainable durum wheat production relies on the constant renewal of agronomic practices, particularly when the application of nutrients becomes a constraint to crop growth and development under drought conditions. Although application of fertilizers is required to enhance overall durum wheat productivity, increasing fertilization may not guarantee a constant grain yield improvement due to variation in nutrient use efficiency of the different varieties. This could cause acidification of the soil, especially if the applied nitrogen is not taken up by the crop during the growing season [32]. Hence, defining appropriate nitrogen fertilizer doses based on a subset of SPAD measurements could be very important for improved yield and environmental sustainability. Although there is a genetic variation component in grain yield, the yield-contributing effect of leaf chlorophyll content was consistent across the tested varieties. The increased yield effect established for optimum leaf chlorophyll content varieties could be because of minimal leaf photochemical damage and optimal absorption of light energy required for maximum photosynthesis [33,34]. This direct association between yield and SPAD values means that SPAD-based nutrient management could play an important role in the efficient management of nitrogen based-fertilizers [17].

Chlorophyll concentration was sensitive to nitrogen supply and varietal response, particularly under drought stress conditions. Under such scenarios, monitoring chlorophyll content in durum wheat leaves could sense a slight nitrogen deficiency which allows early intervention to correct this issue before the yield potential is significantly affected. Thus, managing nitrogen-containing fertilizers based on crop needs as guided by chlorophyll meter readings can maintain optimal crop nitrogen status [35]. Across the tested durum wheat varieties and the applied inputs, relatively uniform SPAD readings with the enhanced qualitative and quantitative traits clearly indicate a strong association among grain yield, protein concentration, and early season SPAD readings. This strong and positive correlation under drought conditions further suggests the use and importance of chlorophyll meters in determining in situ nitrogen status and screening of stress-tolerant wheat varieties [36]. Therefore, a chlorophyll meter could be a viable site-specific and complementary agricultural tool to the traditional plant tissue measurements of leaf nitrogen and chlorophyll status.

A slight change in nitrogen application dose could cause a substantial change in SPAD measurements. This could be due to the fact that plants often invest about 30% of their leaf nitrogen in thylakoid components such as chlorophylls [37]. Higher leaf nitrogen status could promote the accumulation of higher chlorophyll content in the leaves and consequently improve the absorption and decrease reflectance of the red spectrum [38]. Low leaf chlorophyll content was confirmed in the non-nitrogen treated varieties (Table 1, Table 2, Table 3 and Table 4). When nitrogen fertilization is inadequate to meet crop nutrient demands, there is a significant reduction of chlorophyll content and photosynthetic efficiency, and carbohydrate synthesis could occur, and consequently affect grain yield [39]. A decline in photosynthetic efficiency under drought conditions could cause a disparity between absorption and utilization of light energy in the carbon assimilation process which releases more electrons, and triggering the production of reactive oxygen species (ROS) [40]. When the ROS generation is not quenched by antioxidants in a timely manner, this process could further impair plant growth through cell membrane damage, effects on DNA and RNA enzymes, and oxidative stress [41]. Although genetic variability affected the SPAD values, the tested durum wheat varieties had values above the optimal SPAD threshold level (41.8) for wheat [18]. Hence, maintenance of photosynthetic pigments could be a potential indicator of drought tolerance of crops [42]. A greater genetic difference in leaf chlorophyll content has been previously reported [43].

An exogenous application of sulfur-based fertilizers also improved the yield and grain protein content of the Tamadur variety. This means that application of sulfur-containing fertilizers at the flag leaf stage could better maintain the grain yield of drought-sensitive varieties compared to the tolerant varieties, which is consistent with the findings from Lee et al. [23]. A significant difference in the varieties’ response to nutrient application could be associated with uptake efficiency and interaction of the applied inputs with other metal ions in different physiological contexts. It is worth mentioning that restriction in sulfur acquisition could be a cause for reduced growth and grain yield, particularly in resource scarce environments [19]. This restriction in SO_4_^2−^ availability due to drought stress may further restrict nitrate uptake because of limited CO_2_ fixation and decreased flux of SO_4_^2−^ into cysteine, consequently affecting grain yield [17]. The results universally hinted at the importance of adequate application of sulfur fertilizer, particular under drought conditions and consideration of its interaction with the environment is very important as far as grain yield and environmental sustainability is concerned. However, the application of sulfur fertilizer did prevent the Tamadur variety from producing the maximum grain yield, since sulfur moderate the serious drought effects on grain yield and enable the varieties to withstand the stressors. This effect could be due to its role in enhancing the photosynthetic rate and stimulating translocation of photosynthesis towards the sink [44]. Application of sulfur-based fertilizer can therefore be a useful and recommended agronomic practice under water-limited environments [45]. Hence, as an individual agronomic approach may not hit the required target, integrated nutrient supply is very important.

Wheat grain yield is a polygenic trait, mostly resulting from variation in phenotypic features of crops, each with divergent genetic landscapes [46]. Enhanced NDVI and LAI values and their association with developmental stages was observed as a potential avenue of achieving maximum grain yield under drought conditions. There could be 1.1 to 2.6 t/ha of grain yield improvement from each 0.1 unit increase in the NDVI value during the season [47]. An increase in NDVI profiles was also observed with adequate fertilization of nitrogen early in the season but not in later stages. Throughout the developmental phases, nitrogen is distributed from older to younger leaves, which may cause discoloration of the older leaves [48] which is reflected in the NDVI values. Higher LAI could be also due to leaf photosynthetic area expansion and carbohydrate production, which later on contributes to biomass and leaf dry-weight production [49]. As an important agronomic trait, spike density was also improved with higher nitrogen application rates, although this effect varied with the genetic landscape and drought tolerance ability of the tested durum wheat varieties, which is consistent with [50]. Würschum et al. [51] has verified that stay-green nature varieties mostly persist as photosynthetically active, which induces production of fertile spikes. A poor number of spikes have been previously reported because of terminal drought stress [52].

The quality of durum wheat is largely dependent on genetic variation and nutrient supply. There was a clear variation in durum wheat varieties according to their yielding ability, which indicates that high yielding varieties could also produce the maximum grain protein content if nutrient supply is not a limiting factor. Nitrogen fertilization has explained a larger proportion of variation than the genotypes under drought conditions, which indicates that low yielding varieties can produce higher grain protein contents in unfertilized conditions, but high yielding varieties could achieve maximum grain protein content under adequate nitrogen supply. There is a well-known positive correlation between leaf chlorophyll content and grain protein content. The observed association between both traits suggests that if varieties could maintain a leaf chlorophyll content of 55 to 65.8, the grain protein content could improve by 10.13%. As the grain protein content of durum wheat is an important end-use quality factor, maintaining about 13% protein concentration in the grain is crucial to improve the quality of cooked pasta [53]. This has been achieved in our study (14.3–16.8%) through zinc, sulfur, and nitrogen fertilization, even though the percentage of grain protein is under the genetic control of the tested varieties (Table 4). Pasta made from higher (<12%) grain protein content wheat has been observed to be more elastic and stronger than pasta made from lower protein content wheat [53,54]. Higher protein content improves over-cooking tolerance, decreases stickiness [32], improves water absorption, and improves the quality of stored loaves [55,56].

## 5. Conclusions

The influence of nitrogen on SPAD readings and durum wheat productivity is highly dynamic. The potential improvement in grain yield and leaf reflectance features depends primarily on the extent of the variability of the nutrient application and nutrient responsiveness of the durum wheat. Co-fertilization of 60 kg/ha of nitrogen with sulfur could improve the drought tolerance of drought-sensitive varieties such as Tamadur. However, in-depth study is required to clarify the changes in SPAD readings in relation to durum wheat developmental features to suggest the optimal sensing strategy, in order to support site- and season-specific fertilizer application. Under drought conditions, improvement in the grain yield of durum wheat seems to be not associated with the number of spikes per square meter but rather to leaf chlorophyll content and NDVI profiles. When water and nutrient supply is not a limiting factor, the varieties that contained a greater leaf chlorophyll content could produce a higher grain yield. Although the chlorophyll meter was influenced by varietal performance, it could be a potential agricultural device to analyze the spatial and temporal variability of durum wheat varieties and to estimate the grain yield under drought conditions. Integrated application of nitrogen and zinc under water-limited environments could improve grain yield of most nutrient-responsive varieties such as Duragold by about 21.3%. The effects on yield due to zinc, nitrogen, and sulfur fertilizers demonstrates the need to include these essential elements in durum wheat cropping systems.

## Figures and Tables

**Figure 1 life-13-00824-f001:**
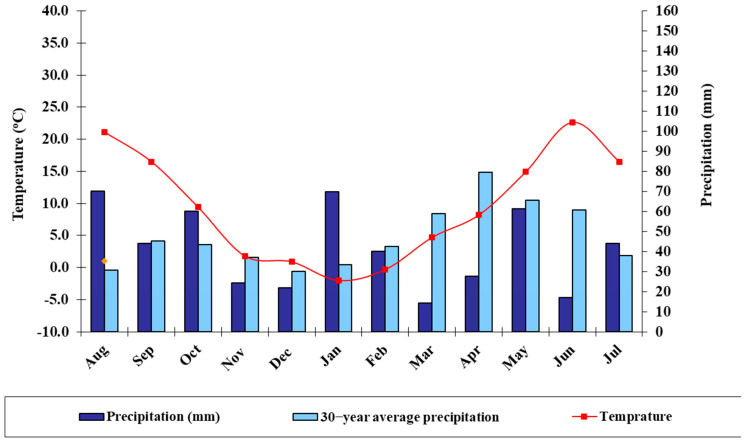
The average daily meteorological data of the Látókép experimental site (i.e., precipitation and temperature) during the 2020/21 cropping season.

**Figure 2 life-13-00824-f002:**
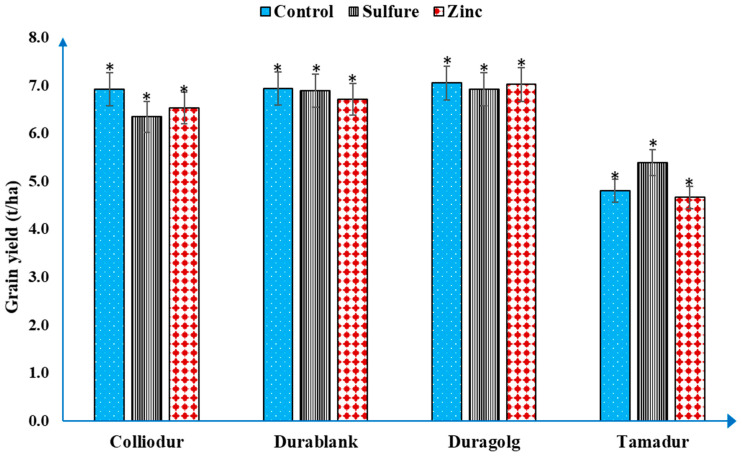
Effects of foliar-based zinc and sulfur fertilization on grain yields of different durum wheat varieties under drought conditions. Bars indicated with single asterisks are statistically significant at 0.05%.

**Figure 3 life-13-00824-f003:**
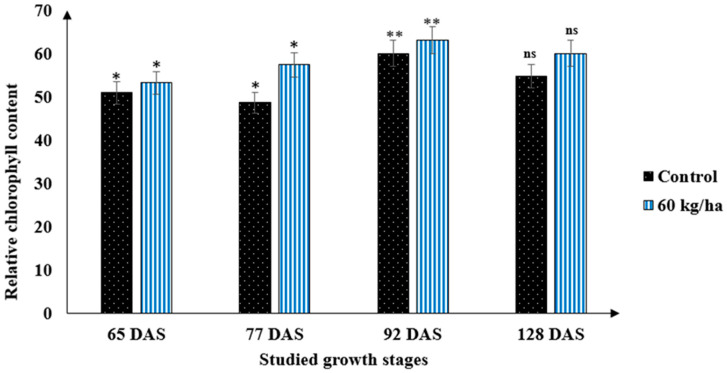
Representations of the dynamics of SPAD readings (chlorophyll content) across different developmental stages in the absence and presence of nitrogen fertilization under drought conditions. Bars indicated as “ns”, is statistically no significant, single asterisks (*) is significant at 0.05% and bars with a double asterisks (**) are statistically significant at 0.01%.

**Figure 4 life-13-00824-f004:**
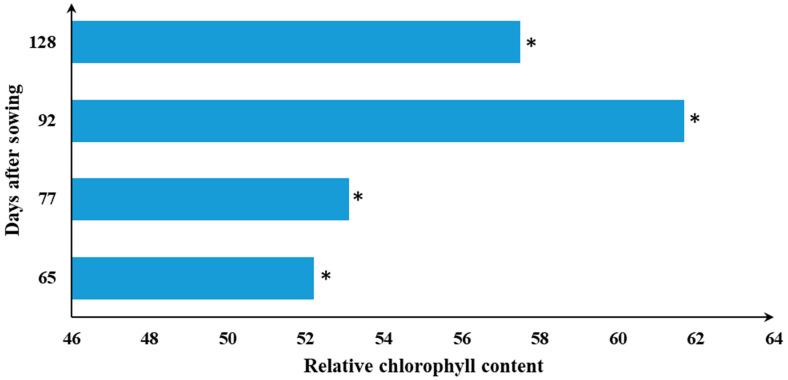
SPAD readings as influenced by developmental stages of durum wheat varieties under drought conditions. Bars indicated in a single asterisks (*) significant at 0.05%.

**Figure 5 life-13-00824-f005:**
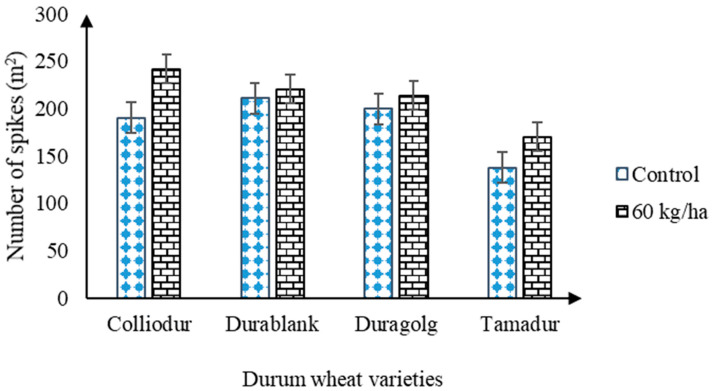
The interplay between nitrogen and varietal differences on spike density (m^2^).

**Figure 6 life-13-00824-f006:**
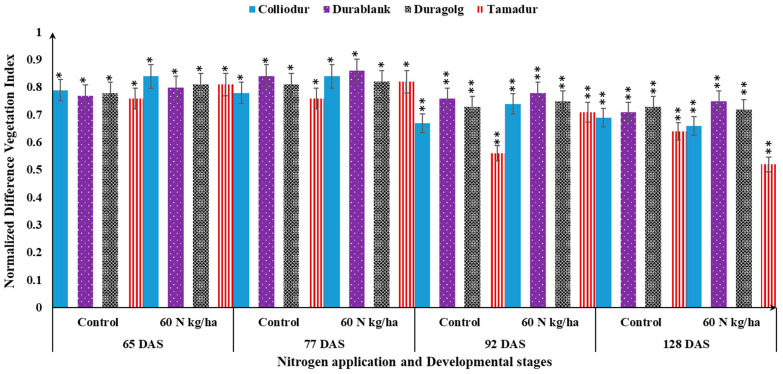
Influence of interaction effects among varieties, nitrogen application, and developmental plasticity on normalized difference vegetation index (NDVI). Bars indicated with a single (*) double asterisks (**) are statistically significant at 0.05% and 0.01%, respectively.

**Figure 7 life-13-00824-f007:**
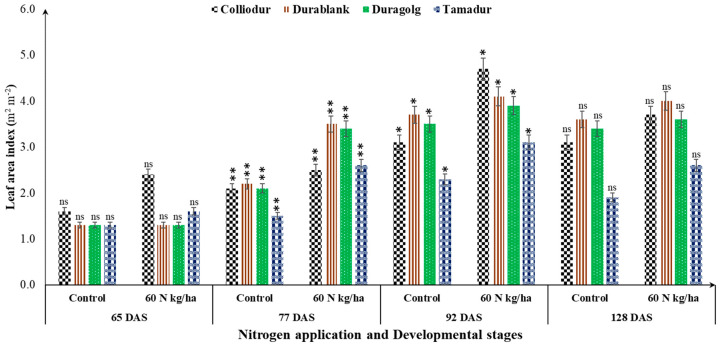
Interaction effects of nitrogen application and varietal differences on leaf area index during progression of developmental stages. Bars indicated as ^ns^, is statistically no significant, single asterisks (*) is significant at 0.05% and bars with a double asterisks (**) are statistically significant at 0.01%.

**Figure 8 life-13-00824-f008:**
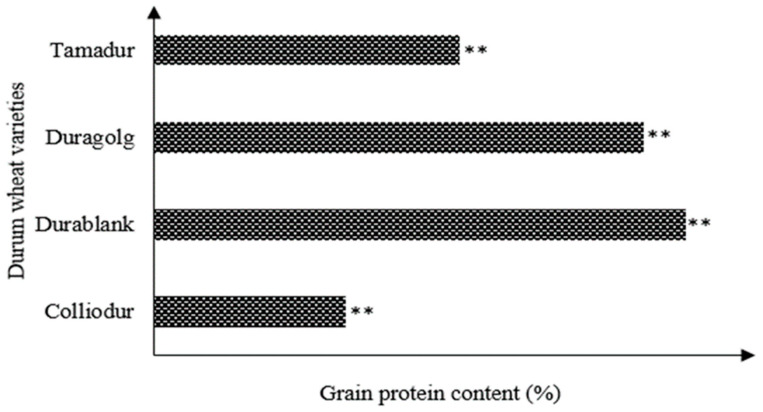
Grain protein content as influenced by genetic divergence in durum wheat varieties. Bars indicated with double asterisks are statistically significant at 0.05% probability level.

**Figure 9 life-13-00824-f009:**
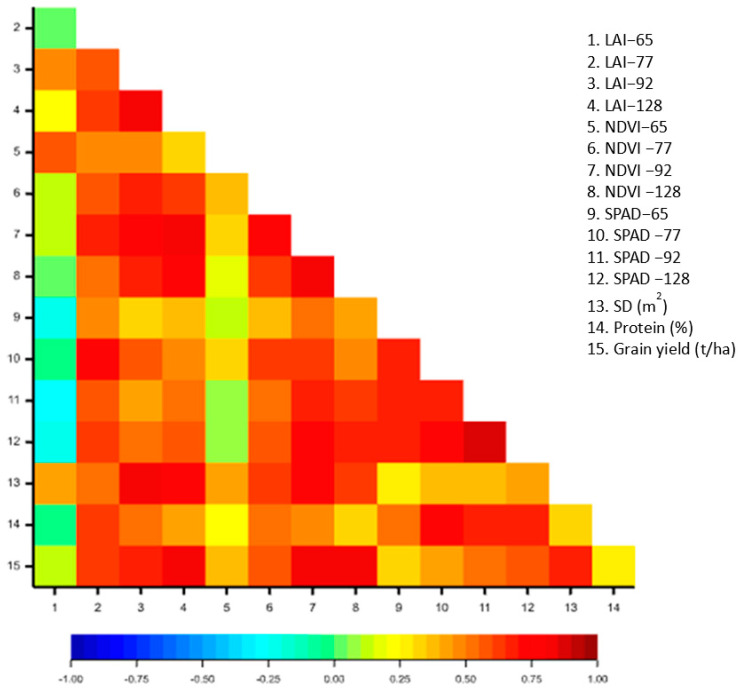
Correlation coefficient among physio-morphological traits, grain yield, and protein content of durum wheat varieties grown under drought conditions.

**Table 1 life-13-00824-t001:** Genetic regulation influences relative chlorophyll content and grain yield under drought conditions.

N Rate (kg/ha)	Varieties	Chlorophyll Measured at Different Stages				SD (m^2^)	GPC (%)	GY (t/ha)
		65 DAS	77 DAS	92 DAS	128 DAS			
Control	Colliodur	49.0	49.0	56.8	52.3	216	14.6	6.61
	Durablank	54.3	56.6	65.3	62.7	216	16.6	6.85
	Duragold	53.6	55.0	65.9	61.6	206	16.3	7.00
	Tamadur	51.8	51.8	58.8	53.5	154	15.3	4.95
LSD _(0.05)_		1.19	1.03	1.36	1.93	10.3	0.61	0.24
CV (%)		3.40	2.90	3.30	5.0	7.7	2.8	3.0

Key to abbreviations: N, nitrogen; SD (m^2^), spike density; DAS, days after sowing; GPC (%); grain protein content; GY (t/ha), grain yield; LSD _0.05_, least significant difference; CV (%), coefficient of variation.

**Table 2 life-13-00824-t002:** Variation in relative chlorophyll content, grain yield, and spike density (m^2^) and protein content influenced by variety × nitrogen interaction during different developmental stages.

N Rate (kg/ha)	Varieties	Chlorophyll Measured at Different Stages				SD (m^2^)	GPC (%)	GY (t/ha)
		65 DAS	77 DAS	92 DAS	128 DAS			
Control	Colliodur	47.2	43.0	54.9	48.6	190.3	13.1	6.51
	Durablank	53.8	52.8	64.7	61.3	211.2	15.8	6.80
	Duragold	53.3	51.7	64.9	59.8	200.2	15.4	6.53
	Tamadur	49.8	47.3	56.0	50.1	137.9	15.2	4.01
60 kg/ha	Colliodur	50.7	55.0	58.7	55.9	241.9	16.2	6.70
	Durablank	54.9	60.4	65.8	64.2	220.8	17.4	6.90
	Duragold	54.0	58.3	66.9	63.3	212.7	17.3	7.45
	Tamadur	53.8	56.4	61.6	56.9	170.3	15.3	5.89
LSD _(0.05)_		2.34	1.36	3.69	2.99	14.52	0.72	0.96
CV (%)		3.40	2.90	3.30	5.00	7.70	2.80	3.00

**Table 3 life-13-00824-t003:** Influence of varietal differences on SPAD readings for different growth stages, grain yield, and protein content.

Varieties	Chlorophyll Measured at Different Stages				SD (m^2^)	GPC (%)	GY (t/ha)
	65 DAS	77 DAS	92 DAS	128 DAS			
Colliodur	49.0	49.0	56.8	52.3	216	14.6	6.61
Durablank	54.3	56.6	65.3	62.7	216	16.6	6.85
Duragold	53.6	55.0	65.9	61.6	206	16.3	7.00
Tamadur	51.8	51.8	58.8	53.5	154	15.3	4.95
LSD _(0.05)_	1.19	1.03	1.36	1.93	10.3	0.61	0.24
CV (%)	3.40	2.90	3.30	5.0	7.7	2.8	3.0

**Table 4 life-13-00824-t004:** A summary of significance for variety, nitrogen rate, zinc, and sulfur interactions on selected physiological traits of durum wheat at different growth stages.

Nutrients	Varieties	Chlorophyll Measured at Different Stages				GPC (%)	GY (t/ha)
		SPAD 65	SPAD 77	SPAD 95	SPAD 128		
Control	Colliodur	49.8	49.4	57.4	52.6	14.3	6.9
	Durablank	53.4	55.8	65.6	62.1	15.3	6.9
	Duragold	53.3	54.1	66.8	62.3	14.3	7.1
	Tamadur	51.0	52.8	56.9	52.7	16.6	4.8
	Colliodur	47.9	47.7	56.4	52.1	16.5	6.4
	Durablank	54.3	56.4	64.3	62.5	16.8	6.9
Sulfur	Duragold	55.0	55.3	65.9	61.1	16.4	6.9
	Tamadur	52.5	51.2	59.4	54.9	16.3	5.4
Zinc	Colliodur	49.2	50.0	56.6	52.1	16.4	6.5
	Durablank	55.3	57.5	66.0	63.6	15.2	6.7
	Duragold	52.7	55.6	65.0	61.4	15.1	7.0
	Tamadur	52.0	51.6	60.1	52.9	15.7	4.7
LSD _(0.05)_		2.1	ns	2.5	3.12	0.7	0.4
CV (%)		3.4	2.9	3.3	5.0	3.1	5.5

Key to abbreviations: SPAD, Soil Plant Analysis Development (SPAD-502 chlorophyll meter).

## Data Availability

The data used to support the findings of this study are available from the corresponding author upon request.

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
