# Peer review of "Chlorophyll Meter: A Precision Agricultural Decision-Making Tool for Nutrient Supply in Durum Wheat (Triticum turgidum L.) Cultivation under Drought Conditions"

_life, 2023, doi:10.3390/life13030824_

Round 1

Reviewer 1 Report

The following points should be improved in the manuscript:

Major points:

1. English quality of the whole manuscript should be checked.

2. The authors should clearly define their model for data analysis and its statistical assumption. All the factors included in the model should be explained. 

3. The authors should respect the logic order of tables and figures and ensure their citation in the text to be in the correct sequence. 

4.The authors should avoid the redundant information in the results section.

5. The authors should provide all the figures with high resolution.

Other points:

Line 120: The relative air humidity is not included in the graph of figure 1.

Line 122: The authors should provide more details about the genotypes used in the study.

Line 123-124: The authors should explain the two levels of nitrogen application. It is not clear if the “standard check” treatment is without nitrogen fertilization or with a certain (unknown) quantity of nitrogen. Please clarify this.

Line 181: The authors should provide the details of the analysis of variance.

Line 181: Please cite the reference of GenStat software

Line 203-204: Table 1: Please provide the meaning of the abbreviations.

Line 270: You mean Figure 4? Please correct.

Line 380: Do you mean section 3.8.? Please correct.

Author Response

Dear reviewer (s)

We are very happy to have received a positive evaluation, and we would like to express our appreciation to the reviewer (s) for the thoughtful comments and helpful suggestions. It was your valuable and insightful comments that led to possible improvements in the current version. The authors have considered the comments and tried our best to address every one of them and the authors welcome further constructive comments if any. We provided detailed, point-by-point responses to the reviewer's comments, attached herewith in the MS word file.

Reviewer 2 Report

The present manuscript entitled 'Chlorophyll meter: A precision agricultural decision-making tool for nutrient supply in durum wheat (Triticum turgidum L.)  cultivation under drought condition' addresses adequately the objectives, the methodology, the results and the discussion. 

There are some spelling and grammar mistakes along the document. I tried to correct them in the annotated version I attach. To my opinion, once these errors are corrected, the article can be published in Life.

Best

Author Response

Dear reviewer (s)

We are thrilled to receive a positive evaluation, and we would like to express our appreciation to the reviewer (s) for the thoughtful comments and helpful suggestions. It was your valuable and insightful comments that led to improvements in the current version. The authors have considered the comments and tried our best to address every one of them. The authors welcome further constructive comments if any.

We gave our detailed, point-by-point responses to the reviewer comments below, whereas we marked the corresponding revisions in colored text in the manuscript file.

With Kind regards,

Anteneh A.

Reviewer 3 Report

Dear author

In a general view, the article has an acceptable connection with the special issue. From the point of view of the quality of the figures and structure of the article, it seems to be of interest. But there are some concerns that need to be addressed. Therefore, my opinion is Minor revision.

In the abstract of the article, it is better to refer to the quantitative results. 

The introduction is rather long. The main concepts related to the research are reviewed and other content can be removed.

What parameters did the field measurements include and at what times?

Where did the Varieties come from? (Origin)

Analysis of variance results for the chlorophyll and other traits like protein Needed.

The conclusion section has been able to clearly show the summary and objectives of the research. You should also consider the managerial implications and future research.

Author Response

Dear reviewer,

We are very happy to have received a positive evaluation, and we would like to express our appreciation to you and the reviewers for the thoughtful comments and helpful suggestions. It was your valuable and insightful comments that led to possible improvements in the current version. The authors have carefully considered the comments and tried our best to address every one of them. The authors welcome further constructive comments if any.

We have given our detailed, point-by-point responses to the reviewer's comments attached herewith, and marked the corresponding revisions in colored (red)/track change text in the manuscript file.  

Reviewer 4 Report

Try to change the title. Also, your figure labelling and the reference of figures need consideration. Results are too lengthy. In addition, conclusions should be 3-4 lines maximum. I have made several comments  for your reference. In my opinion it needs a major revision. 

PS, such type of studies are considered normally with a two year data presentation. 

Best regards,

Author Response

Dear reviewer, 

We are very happy to have received a positive evaluation, and we would like to express our appreciation to you and the reviewers for the thoughtful comments and helpful suggestions. It was your valuable and insightful comments that led to possible improvements in the current version. The authors have carefully considered the comments and tried our best to address every one of them. The authors welcome further constructive comments if any.

Our detailed, point-by-point responses to the editorial and reviewer comments are given below, whereas the corresponding revisions are marked in colored (red)/ track changes in the manuscript file. 

With kind regards,

Round 2

Reviewer 4 Report

The authors have addressed all of my comments. Hence, I have nothing more to say.

Best regards,